# People's perceptions of, willingness-to-take preventive remedies and their willingness-to-vaccinate during times of heightened health threats

**Angela Bearth**[ID]*, **Anne Berthold**[ID], **Michael Siegrist**

Consumer Behavior, Institute for Environmental Decisions (IED), ETH Zurich, Zurich, Switzerland

* angela.bearth@hest.ethz.ch

## Abstract

Pandemics, such as the current SARS-CoV-2 pandemic, represents a health threat to humans worldwide. During times of heightened health risks, the public's perceptions, and acceptance of evidence-based preventive measures, such as vaccines, is of high relevance. Moreover, people might seek other preventive remedies to protect themselves from getting infected (e.g., herbal remedies, nutritional supplements). A recent study on consumers' preference for naturalness showed that people put more weight on perceived naturalness of a preventive remedy compared to a curative one. This result was attributed to the increased focus on perceived effectiveness as opposed to perceived risk. This raises the question whether the current pandemic would shift people's perceptions from prevention to curing and thus, exhibit a preference for synthetic remedies because they are seen as more effective. The present online experiment (conducted in April 2021) investigated people's perceptions of vaccines and remedies within the context of the current SARS-CoV-2 pandemic. A 2x2 between-subject design with type of remedy (natural vs. synthetic) and salience of SARS-CoV-2 pandemic (high vs. low) was conducted in Switzerland in spring 2021 ($N =$ 452). The data did not provide evidence of a curative mindset for preventive remedies, as the participants exhibited a clear preference for the natural remedy compared to the synthetic remedy. Our study stresses the importance of understanding people's mindsets on how to protect themselves from infection with a virus during an ongoing pandemic to tackle misinformation and vaccine hesitancy.

**Data Availability Statement:** Data are available through ETH Zurich's Research Collection, doi: 10.3929/ethz-b-000527437.

## 1. Introduction

Pandemics represent a health threat to humans worldwide and considering the widespread vaccine hesitancy [e.g., 1–3], social science has gained an interest in people's willingness-to-vaccinate and predictive factors of vaccine uptake. Aside from vaccines, people might also seek preventive remedies (e.g., immune system supporting products, household remedies, herbal remedies) to protect themselves from getting infected. Yet, it is unclear how people form their

**Funding:** The author(s) received no specific funding for this work.

**Competing interests:** The authors have declared that no competing interests exist.

opinions regarding these remedies in times of heightened health threat, such as during a pandemic or during flu season [4–6]. The various remedies vary in perceived and scientifically confirmed effectiveness, which was also part of the public discourse throughout the SARS-CoV-2 pandemic (e.g., news media reporting on Echinaforce, recommendation to supplement vitamin D) [6]. Recently, Scott, Rozin (5) found that people prefer remedies advertised as natural for prevention, while they do not exhibit such preferences for curatives. In a series of experiments, the authors [5] could show a consistent effect of naturalness on people's willingness to take a remedy. The authors concluded that lay beliefs regarding perceived safety and potency are the relevant psychological mechanism behind this phenomenon, and that this might also impact responses to newly-developed vaccines [5].

Our study had two goals. The first goal was to investigate people's perceptions of and willingness-to-take a natural or synthetic remedy when a health threat is made salient (vs. not salient). The second goal was to investigate people's perceptions of vaccines and their willingness-to-vaccinate during an acute health threat. In the following two subsections, this study's research questions and hypotheses are presented alongside with the theoretical background and current state of research.

## 2. Theoretical background and research goals

### 2.1 Preventive remedies: State of research, research questions and hypotheses

At the start of the second infection wave of the SARS-CoV-2 pandemic in September 2020, Swiss newspapers broadly covered a story that a plant-based nutritional supplement *echinaforce* could protect against infection with SARS-CoV-2 [7]. This was followed by a surge in purchases throughout the country, with some pharmacies running out of the supplement. However, the conclusions from the original article were misinterpreted, as the study was conducted in the laboratory in an *in vitro* study and no effects were observed *in vivo* [7, 8]. Despite retractions from the newspapers and a public discourse about the limitations of the study, demand remained high for the supplement. This anecdotal story from Switzerland shows that in times of heightened health threat, such as during pandemics, people tend to look for additional ways to protect themselves. Prior studies have shown that health and well-being, as well as the perceived likelihood of getting ill are the most prevalent predictors of intake of remedies and nutritional supplements [9, 10]. Moreover, a large evidence base suggest that natural, plant-based substances are preferred, particularly to prevent illness [5, 11–13]. In their article, Scott, Rozin (5) raised the question under which circumstances, synthetic remedies (e.g., vitamin B12 generated in a laboratory) are preferred compared to natural remedies (e.g., vitamin B12 extracted from soybean plants). In various experiments, they uncovered that naturalness (operationalised as products with naturalness claims, such as "all natural" or "natural relief," or plant-based products) was particularly preferred for preventatives, while for curatives, no such preference could be observed [5]. The mechanism behind this shift is that for curatives, people focus on higher perceived effectiveness, while for preventatives, they focus on perceived risks (e.g., side-effects, costs). Scott and colleagues (2020) devoted a specific section to the implications of their research on remedies in the context of the SARS-CoV-2 pandemic. They raised the question whether pandemics would induce a mindset in the society, which would shift the focus on the higher perceived effectiveness of synthetic remedies compared to natural remedies. Thus, the following research question and hypotheses were investigated within this article:

- R1: Are the type of remedy (natural vs. synthetic), the salience of the SARS-CoV-2 pandemic and prior experiences with remedies related to people's willingness-to-take a remedy?

  ◦ The type of remedy influences people's perceptions of and their willingness-to-take the remedy.

  ◦ The salience of the SARS-CoV-2 pandemic influences people's perceptions of and their willingness-to-take the remedy.

  ◦ The prior experience with remedies (e.g., nutritional supplements, herbal remedies) influences people's perceptions of and their willingness-to-take the remedy.

- R2: Are people's perceptions of a remedy related to their willingness-to-take a natural or synthetic remedy?

  ◦ Perceived naturalness and effectiveness are positively related to people's willingness-to-take the remedy, while perceived risk is negatively related to people's willingness-to-take the remedy.

For R1, we also expected interaction effects between the type of remedy, salience of the SARS-CoV-2 pandemic and prior experience of taking remedies (e.g., that salience might increase the willingness-to-take one type of remedy, while not the other). However, we did not formulate any specific hypotheses due to the lack of solid prior evidence.

## 2.2 mRNA vaccines: State of research, research questions and hypotheses

A variety of individual variables have been suggested that determine whether someone is likely to get vaccinated or will refuse to get vaccinated [1, 14–16]. Among other things, perceived effectiveness, primary risk perception (i.e., regarding getting infected with SARS-CoV-2) and secondary risk perception (i.e., regarding potential consequences of getting vaccinated) were suggested as drivers or barriers of vaccine uptake [15]. Particularly the mRNA vaccines have been proven to effectively protect against infection and severe progressions of SARS-CoV-2 [17–19]. Vaccines are preventive measures that protect someone from getting ill and they are likely to be seen as unnatural, as they are man-made. However, as first formally stated in the article by Scott, Rozin (5), the pandemic and its restrictions on our social lives might induce a mindset, where vaccines are seen as curatives rather than preventatives. Several studies have been published that investigate people's willingness-to-vaccinate and predictive factors before and during the SARS-CoV-2 pandemic [e.g., 3, 15, 20, 21]. The most frequently suggested drivers of vaccine uptake, sometimes investigated in the context of the Protection Motivation Theory are increased personal risk perception and perceived effectiveness [1, 14, 15, 21]. As particularly within a global pandemic, vaccination behaviour is not a solely personal decision, perceptions of health and societal challenges due to the pandemic might also drive vaccine uptake aside from personal risk perception [1]. Vaccine hesitancy regarding the new vaccines against SARS-CoV-2 infections has been attributed to the weighing of costs and benefits, inconvenience and other barriers and doubts regarding the safety of the vaccines due to the quick development and accelerated safety testing [2, 3, 20]. A high risk perception regarding the safety and effectiveness of the vaccine might be associated with people's perception of science and the (un)certainty of scientific findings and thus, higher or lower risk perception [22]. Thus, people that perceive scientific evidence to be more certain will likely perceive less risks than people that perceive scientific evidence as uncertain.

Opinions regarding vaccines are expected to be rather stable and more strongly related to individual variables (i.e., personal risk perception, perceived effectiveness) [1, 14, 16]. Thus, we

did not expect that an experimentally induced salience of SARS-CoV-2 (or the type of remedy seen before) would significantly influence the willingness-to-vaccinate with mRNA vaccines. However, as the questionnaire section for the first aim of this study preceded the questionnaire section for the second aim of the study, potential carry-over effects might emerge. Thus, this will be controlled before investigating the following exploratory research question regarding the mRNA vaccines against SARS-CoV-2:

- R3: How does the perception of mRNA vaccines interact with individual variables (i.e., preference for naturalness, perceived certainty of scientific evidence, pandemic health, and pandemic societal fears) and relate to willingness-to-vaccinate?

## 3. Methodology

### 3.1 Study design and materials

Based on the two separate aims, this study comprised three sections: 1) remedies, 2) mRNA vaccines and 3) individual variables and sociodemographics. To investigate the research questions associated with the first aim of the study, a 2x2 between-subjects design (type of remedy x salience of SARS-CoV-2 pandemic) was applied. The participants were randomly assigned to one of four conditions. Regarding the first independent variable *type of remedy*, the participants were confronted with one of the two types of remedies: 1: *natural remedy* or 2: *synthetic remedy*. The second independent variable was the *salience of SARS-CoV-2 pandemic* with the two expressions 1: *high salience* vs. 2: *low salience*. For the participants in the high salience condition, the questionnaire featured a banner with SARS-CoV-2 pictures above and below the questions, while for the other participants no banner was displayed. Fig 1 presents the visual material used for the independent variables *type of remedy* and *salience of SARS-CoV-2 pandemic*. Naturalness, riskiness, and effectiveness perceptions and willingness-to-take the remedy served as dependent variables. Moreover, various variables served as control variables (e.g., sociodemographics, prior experiences, risk perception).

**3.1.1 First section: Remedies.** This section of the questionnaire was introduced with the following text: "Please imagine that you see an advertisement for the [natural / synthetic] food supplement shown below. Please answer the questions about this food supplement." The text was accompanied by a picture of one of two food supplements (natural / synthetic), which were labelled with the text "to strengthen the immune system" (cf. Fig 1). Next, the scales *perceived naturalness*, *perceived effectiveness* and *perceived risk* were assessed on a five-point semantic differential with three items per scale (cf. Table 1). Cronbach's Alpha was good ($\alpha >$ .80) for all scales for the natural and synthetic remedy. *Willingness-to-take the remedy* was assessed with the following item: "How willing are you to take this food supplement?" with the response options 1: "I would definitely not take this food supplement" to 7: "I would definitely take this food supplement." As prior *experiences with remedies* likely impacts people's perceptions of and willingness-to-take the remedy, this variable was included in the experiment as a control variable. This was assessed by asking participants whether they had previously taken any of a list of remedies, ranging from vitamins (e.g., B, C., D., folic acid, biotin), minerals (e.g., calcium, magnesium, iron), omega-3 fatty acids, probiotics / lactobacilli, amino acids (e.g., lysine, taurine) and Alkaline salts. The participants could also indicate other remedies that they take in an open response field, or they could indicate that they did not take any remedies at all. For the analyses, this variable was recoded into no prior experience with remedies and prior experience with remedies.

**3.1.2 Second section: mRNA vaccines.** The mRNA vaccine section was introduced with the following text: "In Switzerland, currently the two mRNA vaccines from Pfizer/Biontech

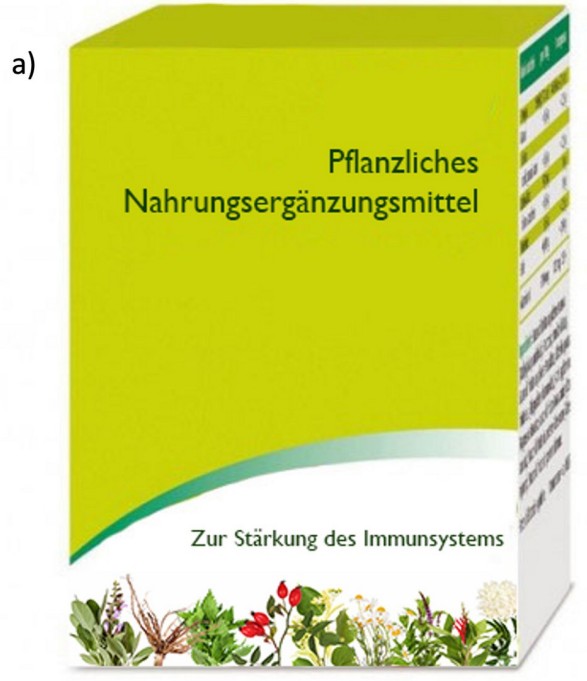

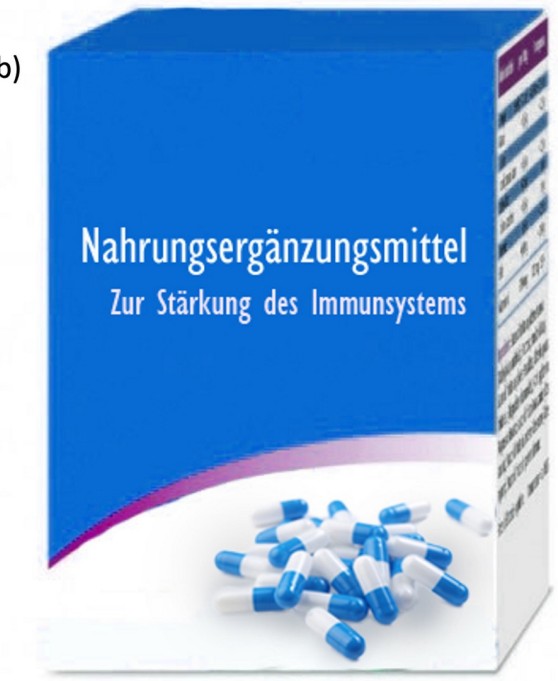

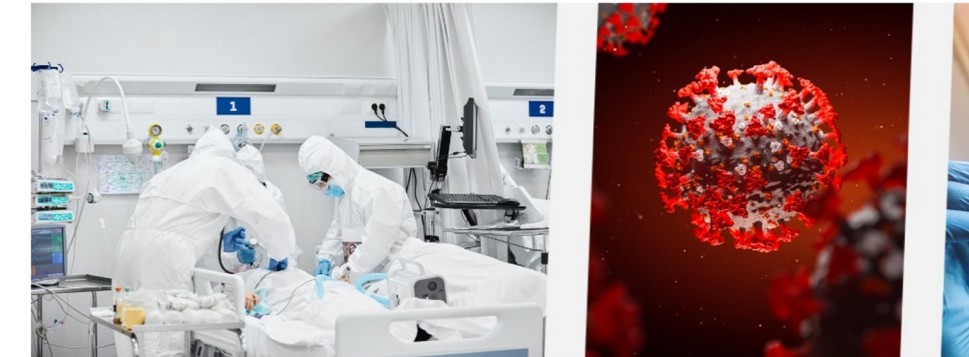
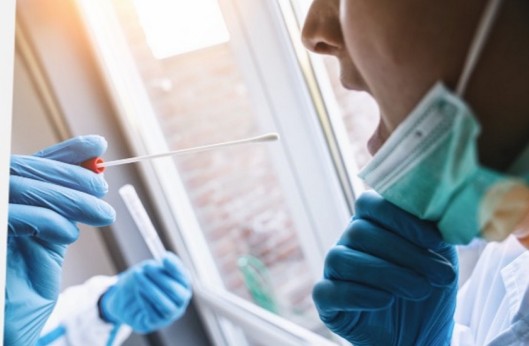

**Fig 1. Visual material used in the study.** a) type of remedy: natural remedy, b) type of remedy: synthetic remedy, c) banner that was placed above and below the study questions. Translation: (Pflanzliches) Nahrungsergänzungsmittel–(natural) food supplement; Zur Stärkung des Immunsystems–to strengthen the immune system.

(BNT162) and Moderna (mRNA-1273) are approved against the corona virus." Next, the *perceived naturalness*, *perceived effectiveness* and *perceived risk* were assessed on the same five-point semantic differential as before (cf. Table 2). Cronbach's Alpha was good ($\alpha > .80$) for all three scales. *Willingness-to-vaccinate* was assessed with the following question "How willing are you to vaccinate with the vaccines currently approved in Switzerland (mRNA from Pfizer/Biontech or Moderna)" and the response options 1 "I would definitely not vaccinate" to 7 "I would definitely vaccinate." The willingness-to-vaccinate was only assessed for the people that

**Table 1. Items for perceived naturalness, effectiveness, and risk scales for natural (n = 225) and synthetic remedy (n = 227).**

| | Natural remedy | | | Synthetic remedy | | |
|---|---|---|---|---|---|---|
| | **M** | **SD** | **$r_i$** | **M** | **SD** | **$r_i$** |
| **Perceived naturalness of remedy** | | | | | | |
| 1: unnatural– 5: natural | 4.0 | 1.0 | .80 | 2.2 | 1.1 | .77 |
| 1: chemical– 5: not chemical | 4.0 | 1.0 | .76 | 2.1 | 1.1 | .76 |
| 1: artificially produced– 5: nature-based | 3.8 | 1.0 | .79 | 2.1 | 1.1 | .75 |
| **Perceived effectiveness of remedy** | | | | | | |
| 1: has no effect– 5: has an effect | 3.2 | 0.9 | .77 | 2.9 | 1.0 | .86 |
| 1: ineffective– 5: effective | 3.2 | 0.8 | .73 | 2.8 | 1.0 | .81 |
| 1: useless– 5: useful | 3.3 | 1.0 | .72 | 2.9 | 1.1 | .82 |
| **Perceived risk of remedy** | | | | | | |
| 1: free of side-effects– 5: has strong side-effects | 2.3 | 0.8 | .63 | 2.8 | 0.9 | .69 |
| 1: harmless– 5: harmful | 2.2 | 0.9 | .72 | 2.9 | 1.0 | .77 |
| 1: safe– 5: dangerous | 2.2 | 1.0 | .77 | 2.9 | 1.0 | .78 |

M, mean; SD, standard deviation; $r_i$, corrected item-scale correlation.

had not yet received the vaccine yet. This screening was conducted by asking the participants if they were a) fully vaccinated (1st and 2nd dose of the vaccine), b) partly vaccinated (1st dose of the vaccine) or c) not vaccinated.

**3.1.3 Third section: Sociodemographics and individual variables.** A series of sociodemographics and individual variables were assessed in this third section.

First, the participants' *gender*, *age* and educational level (*education*) were assessed. The participants were asked whether they work in the medical field (*job*, e.g., pharmacy, hospital, medical practice). Also, participants were asked to rate whether they saw themselves as part of the risk group (*subjective risk group*, response options: yes, certainly; yes, rather; no, rather not; no, not at all; do not know). The responses were recoded into participants that considered themselves part of the risk group and those that did not. To measure *objective*

**Table 2. Items for perceived naturalness, effectiveness, and risk scales for mRNA vaccine (N = 452).**

| | mRNA vaccine | | |
|---|---|---|---|
| | **M** | **SD** | **$r_i$** |
| **Perceived naturalness of mRNA vaccine** | | | |
| 1: unnatural– 5: natural | 2.0 | 1.0 | .73 |
| 1: chemical– 5: not chemical | 1.7 | 0.9 | .63 |
| 1: artificially produced– 5: nature-based | 1.7 | 1.0 | .71 |
| **Perceived effectiveness of mRNA vaccine** | | | |
| 1: has no effect– 5: has an effect | 3.7 | 1.1 | .89 |
| 1: ineffective– 5: effective | 3.7 | 1.1 | .88 |
| 1: useless– 5: useful | 3.8 | 1.1 | .88 |
| **Perceived risk or mRNA vaccine** | | | |
| 1: free of side-effects– 5: has strong side-effects | 3.3 | 1.0 | .72 |
| 1: harmless– 5: harmful | 3.0 | 1.0 | .80 |
| 1: safe– 5: dangerous | 3.0 | 1.0 | .79 |

M, mean; SD, standard deviation; $r_i$, corrected item-scale correlation.

*risk group*, participants over the age of 65 and with pre-existing conditions (i.e., high blood pressure, diabetes type I or II, cardiovascular disease, chronic respiratory disease, diseases, and therapies that weaken the immune system or cancer) were considered being part of the objective risk group. The remaining participants were being considered not part of the objective risk group. Next, people's *prior experience with vaccines* was measured with a series of questions on previous vaccinations (i.e., flu, booster vaccinations, against tick-borne encephalitis). Responses were recoded into no prior experience and prior experience with vaccines.

The individual measures were mostly taken from previous literature. Table 3 presents an overview of the included items for each measure. For *preference for naturalness* a short three-item scale was used [23, 24]. Cronbach's alpha was good with α = .79. *Perceived certainty of scientific evidence* was measured with the objective sub-scale of Retzbach, Otto (22) with a good Cronbach's alpha of α = .79. Both the scales for *pandemic health fears* (α = .92) and *pandemic societal fears* (α = .81) were taken from studies conducted in Switzerland during the SARS--CoV-2 pandemic in 2020 [25, 26]. Two types of *subjective knowledge* were assessed by asking "What do you think: How much do you know about a) vaccines in general and b) mRNA vaccines against the corona virus (SARS-CoV-2)" with the response options 1: "little knowledge" to 7: "a great deal of knowledge."

**Table 3. Items of the individual measures (N = 452).**

|  | *M* | *SD* | $r_i$ |
|---|---|---|---|
| **Preference for naturalness** 1: not important at all– 4: very important |  |  |  |
| It is important to me that the food I eat on a typical day . . . |  |  |  |
| . . . contains no additives. | 3.0 | 0.8 | .61 |
| . . . contains natural ingredients. | 3.3 | 0.7 | .60 |
| . . . contains no artificial ingredients. | 3.1 | 0.8 | .71 |
| **Perceived certainty of scientific evidence** 1: do not agree at all– 7: strongly agree |  |  |  |
| Scientific facts, if carefully examined, are valid for all times. | 2.8 | 1.5 | .47 |
| What has been published in a prestigious scientific journal can be seen as proven. | 3.6 | 1.4 | .60 |
| Scientific predictions are certain when they are well-founded. | 3.7 | 1.4 | .60 |
| Results from an experiment can be seen as proof. | 3.7 | 1.4 | .66 |
| If scientists have worked carefully, their results can be seen as certain. | 4.2 | 1.4 | .54 |
| **Pandemic health fears** 1: do not agree at all– 7: strongly agree |  |  |  |
| I am afraid that. . . |  |  |  |
| . . . I will get infected. | 3.4 | 1.7 | .77 |
| . . . someone in my family or friends will get infected. | 4.1 | 1.7 | .84 |
| . . . there will be deaths in my social circle. | 3.9 | 1.7 | .82 |
| . . . there will be many fatalities in Switzerland. | 3.7 | 1.7 | .79 |
| . . . the healthcare system will be overwhelmed. | 4.4 | 1.7 | .71 |
| **Pandemic societal fears** 1: do not agree at all– 7: strongly agree |  |  |  |
| I am afraid of. . . |  |  |  |
| . . . the personal consequences of an economic crisis in Switzerland (e.g., losing my job, to be on short time, wage cuts) | 4.0 | 1.8 | .59 |
| . . . the societal consequences of an economic crisis in Switzerland (e.g., high unemployment rates, lower wages, falling market price) | 4.8 | 1.6 | .69 |
| . . . changes to the Swiss society (e.g., increasing egoism, lack of solidarity) | 4.7 | 1.6 | .58 |
| . . . crises and conflicts worldwide. | 4.8 | 1.5 | .67 |

M, mean; SD, standard deviation; $r_i$, corrected item-scale correlation.

### 3.2 Data collection and analysis

Prior to data collection, ethical approval was sought by the universities' ethics committee (EK 2021-N-43). Recruiting of this study's participants was supported by a market research company (respondi AG). The required total sample size was determined with an a prior power analysis in G*Power 3 (Cohen's $d$ = .43; β-1: .99, α = .05) [27]. Quota sampling was applied to ensure heterogeneity in gender (50:50) and age of participants (five age groups with equal distribution). The participants were incentivised financially by the market research company to participate. The data was collected anonymised, and all participants provided informed consent before participating. At the end of the questionnaire, all participants were debriefed and thanked.

### 3.3 Data analysis

All data analyses were conducted in SPSS Version 26 [28]. All multi-item scales were subjected to a scale analysis including Exploratory Factor Analysis and reliability analysis using Cronbach's alpha. For R1, ANOVAs were conducted with willingness-to-take remedy as dependent variable and type of remedy, salience of the SARS-CoV-2 pandemic and prior experience with remedies as independent variables. For R2, Pearson correlation coefficients were calculated for the involved variables. For R3, Pearson correlation coefficients were calculated, and a stepwise linear regression analysis was conducted with willingness-to-vaccinate as dependent variable. The perceptions of the mRNA were added in the second step, as they were assumed to be more proximal variables to willingness-to-vaccinate than the sociodemographics and individual variables. Pearson correlation coefficients present the direction (positive vs. negative) and strength of the relationship between two variables. The closer a coefficient is to 1.00, the stronger is the relationship between the variables. A p-value $p < .05$ is considered statistically significant.

## 4. Results

### 4.1 Sample description

In total, $N$ = 486 participants filled out the questionnaire. During data cleaning, $n$ = 34 speeders were identified that spent less than half the median duration on filling out the survey, and were subsequently, removed from the final sample. The final sample comprises $N$ = 452 participants (221 males (49%), 229 females (51%), 2 participants indicating 'other' (<1%); $M_{age}$ = 46, $SD_{age}$ = 15, age range: 18 to 74 years). The sample was heterogenous in terms of educational level with $n$ = 329 indicating low levels of education (73%, e.g., compulsory school, high school, vocational school) and $n$ = 123 indicating a university degree (27%, e.g., BSc, MSc, PhD). Sex, age, and educational level were distributed homogenously across the four conditions, suggesting that the randomisation of participants into four conditions was successful (see S1 Table).

### 4.2 Perception of and willingness-to-take remedy

The subsequent section focuses on the first research question (R1). Three variables assessed the perception of the remedies, and one variable assessed the willingness-to-take the remedy. Table 4 presents the descriptive results (means, 95% confidence intervals) for the included variables. Fig 2 presents a visualisation of the willingness-to-take the remedy separated by condition. The same visualisation for the perception of the remedies can be found in the (S1 Fig).

For perceived naturalness, a main effect ($\eta^2$ = .49) of the type of remedy was observed ($F$(1, 444) = 427.01, $p < .001$). The natural remedy was perceived as significantly more natural than

**Table 4. Perception of and willingness-to-take remedies, separated by experimental condition (type of remedy, salience of SARS-CoV-2 pandemic) and prior experience with remedies.**

| | Prior experience with remedies | | | | no prior experience with remedies | | | |
|---|---|---|---|---|---|---|---|---|
| | high salience | | low salience | | high salience | | low salience | |
| | natural remedy | synthetic remedy | natural remedy | synthetic remedy | natural remedy | synthetic remedy | natural remedy | synthetic remedy |
| | M [95% CI] | M [95% CI] | M [95% CI] | M [95% CI] | M [95% CI] | M [95% CI] | M [95% CI] | M [95% CI] |
| | n = 71 | n = 60 | n = 63 | n = 76 | n = 46 | n = 53 | n = 91 | n = 38 |
| **Perceived naturalness** 1: low– 5: high | 3.9 [3.7, 4.1] | 2.2 [1.9, 2.4] | 4.2 [4.0, 4.5] | 2.3 [2.1, 2.5] | 3.8 [3.5, 4.1] | 1.8 [1.6, 2.1] | 3.9 [3.6, 4.1] | 2.1 [1.8, 2.4] |
| **Perceived effectiveness** 1: low– 5: high | 3.4 [3.2, 3.6] | 3.1 [2.9, 3.3] | 3.5 [3.3, 3.7] | 3.1 [2.9, 3.2] | 3.0 [2.8, 3.3] | 2.4 [2.2, 2.6] | 2.9 [2.6, 3.1] | 2.8 [2.5, 3.1] |
| **Perceived risk** 1: low– 5: high | 2.2 [2.0, 2.4] | 2.9 [2.7, 3.1] | 2.0 [1.8, 2.2] | 2.7 [2.5, 2.9] | 2.3 [2.0, 2.5] | 3.1 [2.9, 3.4] | 2.4 [2.2, 2.7] | 2.8 [2.5, 3.0] |
| **Willingness-to-take** 1: low– 5: high | 4.4 [4.1, 4.8] | 3.5 [3.1, 3.8] | 4.7 [4.4, 5.1] | 3.4 [3.1, 3.8] | 3.9 [3.5, 4.4] | 2.3 [1.9, 2.7] | 3.1 [2.6, 3.5] | 2.7 [2.2, 3.1] |

M, mean; 95% CI, 95% confidence interval; n, sample size per condition.

the synthetic remedy. There was an effect ($\eta^2 = .01$) of the salience of the SARS-CoV-2 pandemic ($F(1, 444) = 5.26$, $p = .022$). Under the condition "high salience," both remedies were perceived as less natural. There was an effect ($\eta^2 = .02$) of the prior experience with remedies ($F(1, 444) = 7.93$, $p = .005$). Participants that had prior experience with remedies perceived them as more natural than participants with no prior experience. None of the interaction effects were found to be significant for perceived naturalness ($F < 1.37$, $p > .243$, $\eta^2 < .003$).

For perceived effectiveness, a three-way interaction effect ($\eta^2 = .01$) was observed ($F(1, 444) = 4.39$, $p = .037$). Participants without prior experience with remedies perceived the synthetic remedy as less effective under the condition of high vs. low salience of the SARS-CoV-2 pandemic, ($\eta^2 = .01$; $F(1, 444) = 4.66$, $p = .031$. No other interaction effects were uncovered ($F < 1.19$, $p > .275$, $\eta^2 < .003$). Further, we found a main effect of the type of remedy ($\eta^2 = .04$; $F(1, 444) = 18.75$, $p < .001$) indicating that the natural remedy was seen as more effective than the synthetic remedy, and a main effect of prior experience with remedies ($\eta^2 = .07$; $F(1, 444) = 34.87$, $p < .001$) indicating that participants with prior experience perceived higher effectiveness than participants without prior experience. There was no main effect of salience of the SARS-CoV-2 pandemic on perceived effectiveness ($\eta^2 = .002$; $F(1, 444) = 0.80$, $p = .371$).

For perceived risks, a main effect ($\eta^2 = .14$) of the type of remedy was observed ($F(1, 444) = 70.13$, $p < .001$). The synthetic remedy was perceived as riskier than the natural remedy. There was an effect ($\eta^2 = .02$) of the prior experience with remedies ($F(1, 444) = 8.61$, $p = .004$). Participants with prior experience reported less perceived risk than participants without prior experience with remedies. The effect of the salience of the SARS-CoV-2 pandemic failed to reach conventional levels of significance, but the perceived riskiness of the remedies tended to be higher under the high vs. low salience condition ($\eta^2 = .008$; $F(1, 444) = 3.46$, $p = .064$). None of the interaction effects were found to reach significance ($F < 3.17$, $p > .076$, $\eta^2 < .007$).

The willingness-to-take remedies was determined by all three variables, as can be seen in the three-way interaction effect ($\eta^2 = .02$; $F(1, 444) = 7.32$, $p = .007$). Participants without prior experience with remedies were more willing to take the natural remedy under the "high salience" than under the "low salience" condition ($\eta^2 = .02$; $F(1, 444) = 7.07$, $p = .008$ (cf. Fig 2). Independent from the type of remedy, the salience manipulation had no significant effect on participants with prior experience with remedies ($F < 1.50$, $p > .221$, $\eta^2 < .003$). The willingness-to-take the natural vs. synthetic remedy was in general significantly higher ($F > 12.97$,

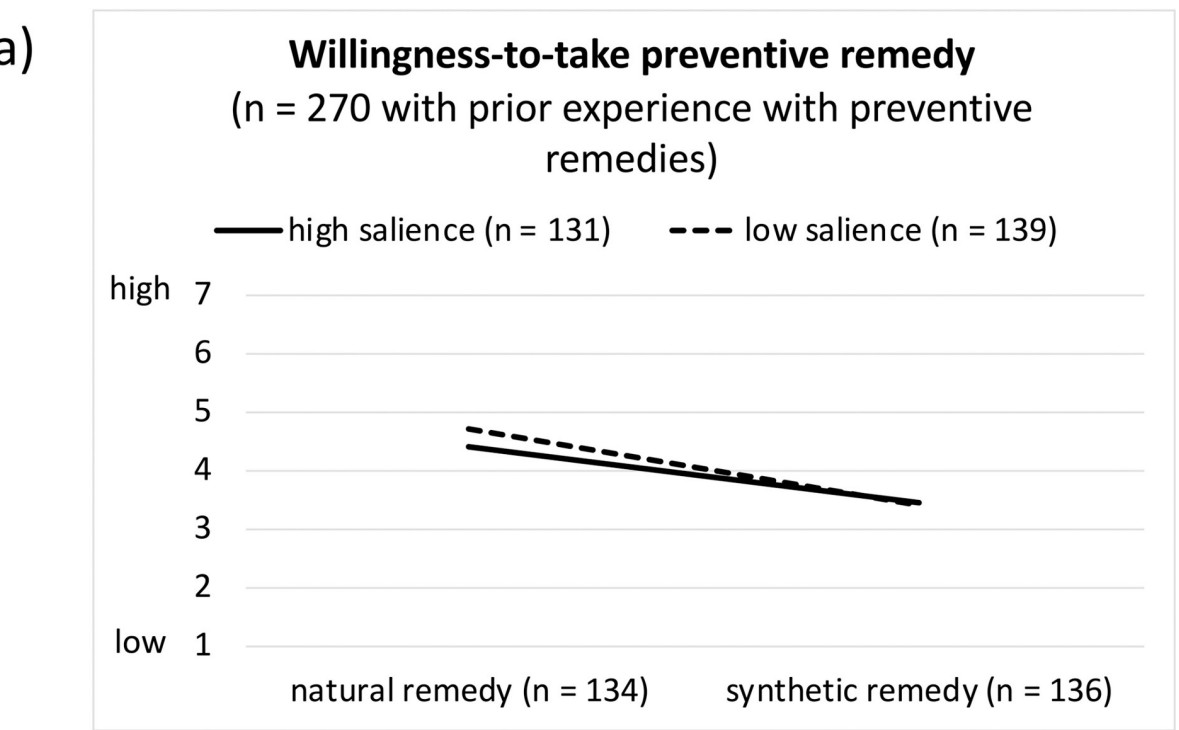

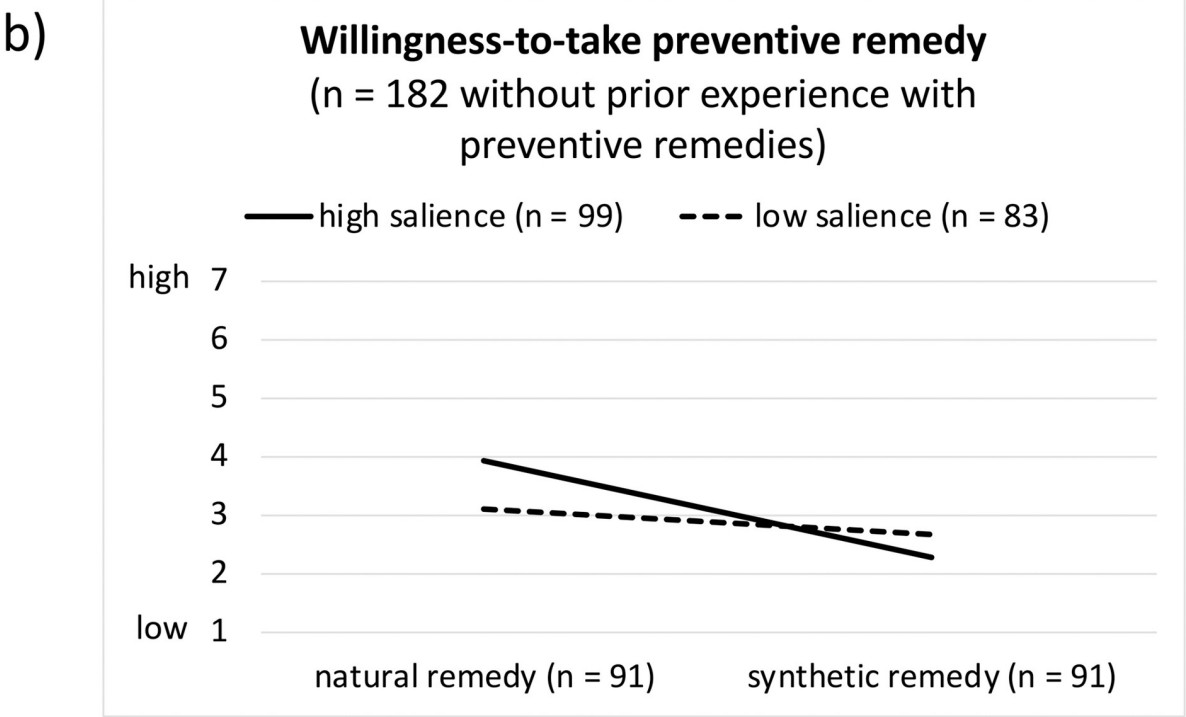

**Fig 2. Willingness-to-take remedy, separated by experimental condition (type of remedy, salience of SARS-CoV-2 pandemic) and prior experience with remedies.** (a) participants with prior experience with remedies, n = 270; b) participants without prior experience with remedies, n = 182).

$p > .001$, $\eta^2 > .028$) except for the participants without prior remedy experience in the low salience condition ($F < 1.66$, $p > .198$, $\eta^2 < .004$). No other interaction effects were uncovered ($F < 2.14$, $p > .144$, $\eta^2 < .005$). Further, a main effect of the type of remedy ($\eta^2 = .11$; $F(1, 444) = 55.51$, $p < .001$) and a main effect of prior experience with remedies ($\eta^2 = .10$; $F(1, 444) = 47.40$, $p < .001$) were observed. These main effects indicate that willingness-to-take the remedy was higher for the natural (vs. synthetic) remedy and for participants with prior experience than for participants without. There was no main effect of salience of the SARS-CoV-2 pandemic on perceived effectiveness ($\eta^2 < .001$; $F(1, 444) = 0.11$, $p = .744$).

Table 5 presents the bivariate correlations between willingness-to-take the natural and synthetic remedy and the perceived naturalness, effectiveness, and risk (R2). The correlation tables are split up by condition. For both types of remedies and for the low and high salience condition, the predicted relationships were found in the predicted directionality. Relationship strengths of the bivariate correlations varied slightly among the four groups. Thus, willingness-to-take the remedy was higher when it was perceived as more natural, more effective, and less risky.

## 4.3 Perception of and willingness-to-vaccinate against SARS-CoV-2

As expected, no carry-over from the first section of the questionnaire was observed, as the type of remedy seen before and the salience of the SARS-CoV-2 pandemic did not significantly impact people's perceptions of the mRNA vaccines (perceived naturalness: $F < 1.39$, $p > .240$, $\eta^2 < .003$; perceived effectiveness: $F < 0.46$, $p > .498$, $\eta^2 < .001$; perceived risks: $F < 1.11$, $p > .293$, $\eta^2 < .002$) and their willingness-to-vaccinate ($F < 0.58$, $p > .446$, $\eta^2 < .002$). Thus, the subsamples were combined to investigate the exploratory research question R3 focusing on the mRNA vaccines against SARS-CoV-2.

In our sample, $n = 45$ (10%) were fully vaccinated, $n = 52$ (12%) were vaccinated with one dose and $n = 355$ (78%) were not yet vaccinated. At the time of the study, the vaccine was not

**Table 5. Bivariate correlations between the perception of and willingness-to-take a natural or synthetic remedy.**

| Natural remedy[a] | | | | |
|---|---|---|---|---|
| | Willingness-to-take remedy | Perceived naturalness | Perceived effectiveness | Perceived risk |
| Willingness-to-take | - | .42*** | .59*** | -.46*** |
| Perceived naturalness | .50*** | - | .44*** | -.56*** |
| Perceived effectiveness | .62*** | .46*** | - | -.29** |
| Perceived risk | -.50*** | -.62*** | -.37*** | - |
| **Synthetic remedy[b]** | | | | |
| | Willingness-to-take remedy | Perceived naturalness | Perceived effectiveness | Perceived risk |
| Willingness-to-take | - | .54*** | .66*** | -.43*** |
| Perceived naturalness | .49*** | - | .44*** | -.35*** |
| Perceived effectiveness | .67*** | .54*** | - | -.42*** |
| Perceived risk | -.53*** | -.51*** | -.56*** | - |

[a] Natural remedy: upper quadrant shows coefficients under "high salience" condition n = 117; lower quadrant shows coefficients under "low salience" condition, n = 108)

[b] Synthetic remedies (upper quadrant shows coefficients under "high salience" condition n = 113; lower quadrant shows coefficients under "low salience" condition, n = 114)

*: p < .05,

**: p < .01,

***: p < .001.

available to the entire population and only the people at risk could get vaccinated (> 65 years, with pre-existing conditions). This explains the low vaccination rate in our sample. The willingness-to-vaccinate with a mRNA vaccine was predominantly high ($M$ = 4.7, $SD$ = 2.3; 1: low– 7: high), with $n$ = 125 (35%) indicating to definitely vaccinate (7). Oppositely, $n$ = 55 (16%) indicated to definitely not vaccinate (1). The remaining participants ($n$ = 175, 49%) provided responses between 2 and 6 on the response scale.

Table 6 presents the bivariate correlations between willingness-to-vaccinate, perceptions of mRNA vaccines and the included individual variables. The strongest relationships between variables were observed for willingness-to-vaccinate and perceived effectiveness (positive) and perceived risk (negative). Furthermore, perceived naturalness, perceived certainty of scientific evidence, pandemic health fears, and subjective knowledge about mRNA vaccines were positively related to willingness-to-vaccinate. Table 7 presents the results of a stepwise linear regression analysis with willingness-to-vaccinate as dependent variable. None of the included sociodemographic variables reached conventional levels of significance, although a trend was found that higher educated participants expressed higher levels of willingness-to-vaccinate ($p$ < .10). Having prior experience with vaccines was also associated with higher willingness-to-vaccinate. In the first step, pandemic health fears were most strongly associated with

**Table 6. Bivariate correlations between willingness-to-vaccinate, perceptions of mRNA vaccines and individual variables (n = 355).**

| | Willingness-to-vaccinate | Perceived naturalness | Perceived effectiveness | Perceived risk | Preference for naturalness | Perceived certainty of scientific evidence | Pandemic health fears | Pandemic societal fears | Subj. knowledge: Vaccines in general | Subj. knowledge: mRNA vaccines |
|---|---|---|---|---|---|---|---|---|---|---|
| Willingness-to-vaccinate | - | | | | | | | | | |
| Perceived naturalness | .33*** | - | | | | | | | | |
| Perceived effectiveness | .72*** | .31*** | - | | | | | | | |
| Perceived risk | -.71*** | -.45*** | -.77*** | - | | | | | | |
| Preference for naturalness | -.11* | -.02 | -.08 | .15** | - | | | | | |
| Perceived certainty of scientific evidence | .30*** | .20*** | .33*** | -.33*** | .05 | - | | | | |
| Pandemic health fears | .39*** | .24*** | .45*** | -.34*** | .08 | .42*** | - | | | |
| Pandemic societal fears | -.03 | .03 | -.03 | .07 | .12* | .19*** | .42*** | - | | |
| Subj. knowledge: Vaccines in general | -.03 | .07 | -.07 | .04 | .17** | .13* | .02 | .15** | - | |
| Subj. knowledge: mRNA vaccines | .11* | .12* | .04 | -.09 | .09 | .09 | .01 | .05 | .69*** | - |

n, sample size without participants that had already been vaccinated at least once.

*: $p$ < .05,

**: $p$ < .01,

***: $p$ < .001.

**Table 7. Stepwise linear regression analysis with willingness-to-vaccinate as dependent variable (n = 332).**

| | B [95% CI] | β | p |
|---|---|---|---|
| **Step 1[1]** | | | |
| Constant | 2.67 [1.17, 4.18] | | .001 |
| Gender (0: male) | -0.28 [-0.70, 0.14] | -.06 | .195 |
| Age | 0.01 [-0.01, 0.03] | .06 | .273 |
| Education (0: low) | 0.46 [-0.01, 0.93] | .09 | .054 |
| Job (0: not in the medical field) | -0.16 [-0.97, 0.65] | .02 | .699 |
| Subjective risk group (0: no) | -0.33 [-0.93, 0.26] | -.06 | .271 |
| Objective risk group (0: no) | -0.09 [-0.67, 0.49] | -.02 | .756 |
| Prior experience with vaccines (0: no) | 1.19 [0.72, 1.67] | .23 | < .001 |
| Preference for naturalness | -0.35 [-0.69, -0.01] | -.10 | .042 |
| Perceived certainty of scientific evidence | 0.39 [0.18, 0.61] | .18 | < .001 |
| Pandemic health fears | 0.64 [0.47, 0.80] | .42 | < .001 |
| Pandemic societal fears | -0.36 [-0.54, -0.18] | -.21 | < .001 |
| Subjective knowledge: Vaccines in general | -0.40 [-0.63, -0.16] | -.22 | .001 |
| Subjective knowledge: mRNA vaccines | 0.34 [0.13, 0.55] | .21 | .001 |
| **Step 2[2]** | | | |
| Constant | 3.81 [1.78, 5.83] | | < .001 |
| Gender (0: male) | -0.24 [-0.56, 0.08] | -.05 | .134 |
| Age | 0.00 [-0.01, 0.01] | -.01 | .891 |
| Education (0: low) | 0.16 [-0.19, 0.52] | .03 | .363 |
| Job (0: not in the medical field) | -0.24 [-0.85, 0.37] | -.03 | .432 |
| Subjective risk group (0: no) | -0.30 [-0.74, 0.15] | -.06 | .190 |
| Objective risk group (0: no) | -0.12 [-0.55, 0.31] | -.03 | .581 |
| Prior experience with vaccines (0: no) | 0.59 [0.22, 0.95] | .11 | .002 |
| Preference for naturalness | -0.17 [-0.43, 0.08] | -.05 | .175 |
| Perceived certainty of scientific evidence | 0.12 [-0.05, 0.28] | .05 | .168 |
| Pandemic health fears | 0.28 [0.15, 0.41] | .18 | < .001 |
| Pandemic societal fears | -0.15 [-0.29, -0.02] | -.09 | .030 |
| Subjective knowledge: Vaccines in general | -0.15 [-0.33, 0.02] | -.09 | .084 |
| Subjective knowledge: mRNA vaccines | 0.15 [-0.01, 0.31] | .09 | .065 |
| Perceived naturalness of mRNA vaccine | 0.08 [-0.13, 0.29] | .03 | .463 |
| Perceived effectiveness of mRNA vaccine | 0.77 [0.54, 1.00] | .36 | < .001 |
| Perceived risk of mRNA vaccine | -0.81 [-1.10, -0.53] | -.32 | < .001 |

Dependent variable: willingness-to-vaccinate.

[1] Step 1, $R^2$ = .34; $F_{(13, 319)}$ = 12.79, p < .001; added: sociodemographic variables, individual variables

2 Step 2, $R^2$ = .64; $F_{(13, 316)}$ = 34.64, p < .001; added: sociodemographic variables, individual variables, and perception of mRNA vaccines

n, sample size without participants that had already been vaccinated at least once; B, unstandardised regression coefficient; 95% CI, 95% confidence interval of B; β, standardised regression coefficient; p, probability of the hypothesis test.

willingness-to-vaccinate: higher pandemic health fears were associated with higher willingness-to-vaccinate. Participants that perceived scientific evidence as certain and those with high subjective knowledge about mRNA vaccines also expressed higher willingness-to-vaccinate. Pandemic societal fears, subjective knowledge about vaccines in general and preferences for naturalness were negatively related to willingness-to-vaccinate. In the second step, perceived effectiveness and risk were most strongly associated to willingness-to-vaccinate. Further, the

relationship between prior experience with vaccines, pandemic health and societal fears and willingness-to-vaccinate remained significant yet diminished in their effect size. Perceived certainty of scientific evidence ($p$ = .168) and the two subjective knowledge items were no longer significantly related to the willingness-to-vaccinate ($p < .10$).

## 5. Discussion

The SARS-CoV-2 pandemic represents a global threat, which has changed the lives of many. As manifold as people's perception of this threat, are their responses and actions taken to protect themselves. While some measures have been shown to effectively protect from infection and severe progressions of infections (i.e., social distancing, wearing a facemask, vaccinating), people might also seek non-evidence-based protection. Investigating the reasons for people's various responses to protective measures might help understanding vaccine hesitancy and to address misinformation. For this reason, this study focused on people's perceptions of evidence-based (i.e., mRNA vaccines) and non-evidence-based preventive measures (i.e., nutritional supplement) against infection with SARS-CoV-2.

### 5.1 Natural vs. synthetic preventive remedies and the salience of the SARS-CoV-2 pandemic

Scott, Rozin (5) found that people prefer "natural" for prevention compared to curing and attributed this to the shifting focus on expected effectiveness instead of expected risks (e.g., side effects, costs). This raised the question whether the ongoing SARS-CoV-2 pandemic induces a focus on perceived effectiveness and thus, increase people's preference of synthetic vs. natural remedies. Accordingly, we conducted an experiment with a 2x2 design to test if the willingness to take a synthetic (vs. natural) remedy will be increased; especially when the SARS-CoV-2 pandemic is made salient (vs. not) via pictures displayed throughout the study on the screen.

Our data did not find consistent evidence of a preference for synthetic preventive remedies, as our participants were more inclined to take the presented natural remedy than the synthetic remedy. Contrary to Scott, Rozin (5), our participants perceived the natural remedy as more effective than the synthetic remedy. This might offer one explanation, why no preference for synthetic remedies was uncovered in the high salience condition. Moreover, participants evaluated the natural remedy as more natural and less risky. The relationships between willingness-to-take and the assessed perceptions of the preventive remedies were in the expected directions in all 2x2 experimental groups: if the remedy was seen as natural, effective, and low in risk, willingness-to-take was higher. It is possible that the missing effect of preferring the synthetic over the natural remedy in times of the pandemic can be explained by the fact that the vaccines were already available at the time of data collection. Thus, an effective solution that protects from an infection with SARS-CoV-2 and severe outcomes of such an infection was existing. Prior to the approval of the vaccines, the results regarding the synthetic remedy might have looked differently.

Against expectations from prior literature [5, 11], the natural remedy was perceived as more effective than the synthetic remedy. This is mostly explained by the uncovered interaction effects for perceived effectiveness and willingness-to-take. Participants without prior experience with nutritional supplements perceived the effectiveness of the synthetic remedy as lower than of the natural remedy and expressed higher willingness-to-take the natural remedy under the 'high (vs. low) salience' condition. Thus, reminding participants that had never taken a preventive remedy before of the current SARS-CoV-2 pandemic, increased their openness towards the natural remedy. As perceived risk was seen as lower for the natural remedy

than the synthetic one, this could indicate the colloquial mindset of "If it does not protect me, it does at least no harm."

These findings have some implications for the communication with the public during times of heightened health threats. During the early stages of the SARS-CoV-2 pandemic, uncertainty about infection routes and effective protective measures was high, which has been suggested to be a fruitful ground for the spread of misinformation [4, 6]. Various remedies were touted as potential miracle products (e.g., plant-based nutritional supplements, taking drugs, vitamins, injecting or ingesting disinfectant) during the pandemic [7, 29, 30]. Particularly vulnerable people (e.g., with preexisting conditions, older people) might be willing to spend substantial amounts of money on bogus remedies and should be protected from misinformation [26, 31].

We did not find evidence for the preference for synthetic preventive remedies, induced by the pandemic, as suggested by Scott, Rozin (5). As discussed above, this could be due to the fact that that the natural remedy was judged to be more effective. An alternative explanation for a failure to find such an effect could also be found in the experimental design. While the use of affective pictures to prime participants or induce salient images and emotions is an established method in the social sciences [e.g., 32, 33], it is not clear whether salience was successfully impacted. Pandemic health and societal fears, which could serve as proxy measures for the salience of the SARS-CoV-2 pandemic were not significantly higher in the "high salience" than in the "low salience" condition (see S2 Table). This is not that surprising, as a high individual stability in these perceived pandemic fears is expected. However, it suggests that, as the pandemic was still ongoing during data collection, all the participants might have thought about SARS-CoV-2 while responding to the questions. Thus, the "low salience" condition likely does not represent truly low salience. This is an important limitation of this study. Future studies, conducted after the general salience of the pandemic recedes, should investigated this point further and include a manipulation check for salience. Similarly, we used fabricated pictures of natural remedies to avoid effects based on recognition and prior experience with actual remedies. We cannot rule out that the findings would have been different for real remedies available on the market although Scott, Rozin (5) found overlaps in their findings for fictitious and actual products.

## 5.2 Willingness-to-vaccinate against SARS-CoV-2

As expected, perceived effectiveness and perceived risk were of high importance for people's willingness-to-vaccinate. As other studies on vaccination uptake have suggested [1, 15, 16, 20], the speed of vaccine development impacted people's perceptions of their safety, which in turn might be one of the relevant factors for hesitancy to get vaccinated. Our study further shows that this perception might be linked to people's views of science. The more people believe that scientific evidence is certain, the higher they perceived the effectiveness to be and the lower was the risk associated with the mRNA vaccines.

While perceived naturalness plays a role in people's views of remedies, such as nutritional supplements, this did not impact people's willingness-to-vaccinated. This is in line with a mindset [5], which places more weight on effectiveness, whereas perceived naturalness of the vaccine is less important. Despite this, people, who prefer naturalness, for example when it comes to food, perceived higher levels of risk associated with the vaccine, which in turn might have decreased their willingness-to-vaccinate.

Another finding suggests that pandemic health fears increase people's willingness-to-vaccinate, while also impacting people's perceptions of a vaccine's effectiveness and risk. This further supports the idea of people relying on a preventative (i.e., the vaccine) as an effective "cure" for

their ongoing pandemic health fears and discounting potential risks linked to the vaccine [5]. Pandemic societal fears on the other hand were associated with lower willingness-to-vaccinate, which is surprising, as the vaccine is publicly discussed as the easiest way to return to normalcy (e.g., opening of shops and restaurants, decreased need for social distancing).

The study revealed an important finding regarding people's subjective knowledge. Namely, participants indicating to know more about vaccines in general, reported lower willingness-to-vaccinate, while higher subjective knowledge about mRNA vaccines was related to higher willingness-to-vaccinate. A possible explanation for this conflicting finding regarding subjective knowledge might be associated with people's feelings of uncertainty and their actual knowledge. First, people that indicate to know more about mRNA vaccines probably also experience lower uncertainty and novelty regarding the vaccine. This might be, but does not have to be, linked to higher objective knowledge about how the vaccine works. Second, people that indicate to know more about vaccines in general might not really know more. In fact, it is even possible that they know less about vaccines- as similar relationships between subjective and objective knowledge and acceptance were found for genetically modified foods: People that indicated to know more about genetically modified food actually achieved lower scores on objective knowledge and also, expressed lower acceptance of genetically modified food [34, 35]. However, it is important to stress that knowledge provision will likely not change people's minds regarding the mRNA vaccine, as vaccine hesitancy is best tackled with communication targeted to individual's needs and by facilitating vaccine access by eliminating barriers [2, 16, 36, 37]. Furthermore, these are just speculations as no measure of objective vaccine knowledge was included in this study. Both potential explanations could be investigated in future studies by including an objective knowledge scale about vaccines in general and mRNA vaccines [e.g., 38].

A limitation that should be noted is that people were excluded from this question that were already vaccinated. As this excludes the participants with the highest willingness-to-vaccinate, the absolute values of willingness-to-vaccinate likely underestimate the willingness-to-vaccinate in our sample. However, we do not expect that this will have a strong effect on our results, as at the time of data collection the vaccine was not yet available to the larger public. Additionally, we would like to add the following two limitation regarding our findings on willingness to vaccinate. First, emerging and prior literature lists various relevant aspects that contribute to vaccine hesitancy (e.g., individual, contextual or vaccine and vaccination-specific issues), which our article did not consider [2, 36, 39]. Second, as the pandemic is ongoing, the factors predicting vaccine hesitancy might change and other psychological mechanisms might be at work. This study measured early perceptions of the emerging vaccines, which might explain the importance of perceived effectiveness and risk. Repeating the study at a later stage might reveal other relevant predictors (e.g., psychological reactance, trust, accessibility).

## 6. Conclusion

This study attempted to experimentally investigate some of the mechanisms that were suggested in [5]. Our findings suggests that people are willing to take natural remedies to protect themselves from infection, independent of the scientific evidence on the effectiveness of such remedies. The prevailing attitude driving this might be the belief that natural remedies might support the immune system and if they do not, at least they do not have any side effects. Aside from this, closer attention should be paid to the various reasons for vaccine hesitancy. As the pandemic is still ongoing worldwide and new mutations threaten our newly gained freedom, it is important to investigate the reasons for people's hesitancy to vaccinate. Our study found that perceived effectiveness and risk play important parts in willingness to vaccinate. Providing information about the vaccine's safety and effectiveness to reduce perceived uncertainty could

be one public health strategy. Aside from this though, it is important to pay attention to the targeted audience, as confidence and trust in science, consumer preferences and fears might be important pre-conditions for the reception of a particular communication about vaccines.

## Supporting information

**S1 Table. Sociodemographics separated by condition and statistical test of similarity in sociodemographics among the four conditions.** N, number of participants; M, mean; SD, standard deviation; [1] due to low case number 'other' were excluded for the statistical test.
(DOCX)

**S2 Table. Independent t-test comparing pandemic health fears for participants in the low and high salience condition.** M, mean; SD, standard deviation; n, sample size per condition; t, t-value; df, degrees of freedom; p, probability of the hypothesis test.
(DOCX)

**S1 Fig. Perception of remedies, separated by experimental condition.** Type of remedy, salience of SARS-CoV-2 pandemic) and prior experience with remedies (a) participants with prior experience with remedies, n = 270; b) participants without prior experience with remedies, n = 182.
(TIF)

## Author Contributions

**Conceptualization:** Angela Bearth, Anne Berthold, Michael Siegrist.

**Data curation:** Angela Bearth.

**Formal analysis:** Angela Bearth, Anne Berthold.

**Investigation:** Angela Bearth, Anne Berthold.

**Methodology:** Angela Bearth, Anne Berthold.

**Project administration:** Angela Bearth.

**Visualization:** Angela Bearth, Anne Berthold.

**Writing – original draft:** Angela Bearth.

**Writing – review & editing:** Anne Berthold, Michael Siegrist.

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
