## [Decision Letter · Decision Letter 0]

10 Dec 2021

PONE-D-21-32339People’s perceptions of, willingness-to-take preventive remedies and their willingness-to-vaccinate during times of heightened health threatsPLOS ONE

Dear Dr. Authors,

Thank you for submitting your manuscript to PLOS ONE. After careful consideration, we feel that it has merit but does not fully meet PLOS ONE’s publication criteria as it currently stands. Therefore, we invite you to submit a revised version of the manuscript that addresses the points raised during the review process.

We look forward to receiving your revised manuscript.

Kind regards,

Kelvin Ian Afrashtehfar, M.Sc., D.D.S.,Dr. med. dent., FRCDC

Academic Editor

PLOS ONE

Journal Requirements:

Additional Editor Comments (if provided):

Dear author(s),

Kindly comply with the reviewers minor revisions.

Thank you.

The Academic Editor

Reviewers' comments:

Reviewer's Responses to Questions

**Comments to the Author**

1. Is the manuscript technically sound, and do the data support the conclusions?

Reviewer #1: Yes

Reviewer #2: Yes

2. Has the statistical analysis been performed appropriately and rigorously? 

Reviewer #1: I Don't Know

Reviewer #2: Yes

3. Have the authors made all data underlying the findings in their manuscript fully available?

Reviewer #1: Yes

Reviewer #2: Yes

4. Is the manuscript presented in an intelligible fashion and written in standard English?

Reviewer #1: No

Reviewer #2: Yes

5. Review Comments to the Author

Reviewer #1: Thank you for the opportunity to provide peer review for the article „People’s perceptions of, willingness-to-take preventive remedies and their willingness-to-vaccinate during times of heightened health threats”.

The well-written article focuses on a sample of 452 individuals in Switzerland and explores their perception of natural or synthetic remedies as well as their willingness to take those remedies when a health threat is made salient. Furthermore, the authors investigated people’s perceptions of vaccines and their willingness to vaccinate against SARS-CoV-2. In the study, the authors tried to identify possible influencing/predicting factors for both the willingness to take remedies as well as for the willingness to vaccinate against SARS-CoV-2. The authors argue that, especially in times of heightened health threats such as the current Covid-19 pandemic, it is important to identify such factors particularly to prevent misinformation and to prevent vaccine hesitancy. The information gained could be used by governments to launch appropriate strategies to increase vaccine acceptability.

Firstly, the study found that even when the actual health threat was made salient the expected curative mind set did not arise and people still preferred natural remedies. Secondly, the authors identified different factors such as expected effectiveness, low risk and perceived naturalness, that were positively influencing on the perception and the willingness to take remedies.

Thirdly, the data shows which factors are associated with people’s willingness to vaccinate.

The following points should be considered:

- The authors present “perceived naturalness of a preventive remedy” and later write about “a curative mindset”. This is confusing and likely inappropriate since all approaches in the context of SARS-CoV2 infection are preventive or supportive at best and to the best or our knowledge not curative as such.

- Please give a definition for type of remedy natural vs. non-natural, e.g. supplement like vitamin D which may be perceived natural by some and not by others.

- Please state what the role of the market research company (respondi AG) was in the context of the presented work.

- Please explain how participants were selected and how representative the population is for the Swiss population.

- The participants were randomly assigned to one of four conditions. Please show the characteristic for these conditions separately (e.g. as supplementary appendix)

- In Table 5 it is not clear which quadrants the authors are referring to and how the results in the table should be interpreted.

- Further explanation how the numbers in Table 6 can be interpreted would be appreciated.

For the editor: I could not adequately address the statistical methods.

Minor points:

- Baseline characteristics that were collected (seen in Table 7) should be mentioned in the methods.

- Figure 1: Description of pictures: 1, 2 and 3 instead of a, b and c.

- Line 209: Both the scales for pandemic health fears (…)

- It is unclear if “unnatural/natural” vs. “chemical/not chemical” or “has no effect/has an effect” vs. “ineffective/effective” evaluate something different in Table 1 and 2.

- Abbreviation missing for “r” in Table 3, “n” in Table 4 and Table 6 and “n”, “B”, “95% CI”, “Beta” and “p” in Table 7.

- It should be stated in the methods which p-value was deemed statistically significant.

- Write number of excluded persons in parenthesis or total number of remaining persons outside the parenthesis to avoid misunderstanding.

- Sentence on lines 467-469 (As our study found the important role of perceived …) is difficult to understand. Consider adjustment.

- The manuscript needs some revision on spelling and punctuation.

In section 5.1 the authors claim that a curative mindset (which means preference for a more effective drug/remedy) could arise if the perceived risk of a health threat is higher. In prior studies the synthetic drugs/remedies were deemed to be more effective and thus a curative mindset let to a preference of the synthetic drug/remedy. In this study the authors discuss that there is no curative mindset based on the fact that the willingness to take the natural remedy is still preferred over the synthetic one. Considering the fact that in this study the perceived effectiveness of the natural remedies was higher than that of the synthetic remedy (Table 1), the conclusion that there is no curative mindset doesn’t seem logical. The result mean that even if there was a shift towards a curative mindset people in this study would still choose the natural remedy, as they perceived it as more effective.

The number of discussed limitations in the study is limited:

Due to the pandemic during the time of data collection, there was probably no real

low-salience group. Perceived current health threat would have been interesting to try to evaluate if there was a difference between the low salience and high salience group.

No objective test evaluating the knowledge about the vaccines was done. Data based on self evaluation.

The study was conducted at an early stage after vaccinations against SARS-CoV-2 was available. This represents the early vaccination hesitancy and maybe not the current situation. A revaluation would be necessary to evaluate the current situation.

Compared to the SAGE-WG Model the authors of this study only include a small number of the numerous mentioned possible determinants of vaccine hesitancy (H. J. Larson et al.; Understanding vaccine hesitancy around vaccines and vaccination from a global perspective, A systematic review. https://doi.org/10.1016/j.vaccine.2014.01.081)

Reviewer #2: Thank you for the opportunity to review the article. This manuscript highlight people's perception regarding natural and synthetic remedies. There are at least two limitations to this manuscript, first the sample size is very small especially during this time of pandemic where literally everyone on the globe is involved, so the denominator actually should be way larger than what assumed to be the targeted population. the second limitation is that there is a time factor in shaping the opinion of people regarding the investigated perspectives, in the beginning of the pandemic people had a lot of misconceptions that were corrected as more information becomes available and reiterated in media.

6. PLOS authors have the option to publish the peer review history of their article (what does this mean?). If published, this will include your full peer review and any attached files.

Reviewer #1: **Yes: **Roman Adam

Reviewer #2: **Yes: **Sameh W. Boktor

---

## [Author Response · Author response to Decision Letter 0]

22 Dec 2021

Response to reviewers added as document

---

## [Editor Report · Decision Letter 1]

17 Jan 2022

People’s perceptions of, willingness-to-take preventive remedies and their willingness-to-vaccinate during times of heightened health threats

PONE-D-21-32339R1

Dear Dr. Authors,

We’re pleased to inform you that your manuscript has been judged scientifically suitable for publication and will be formally accepted for publication once it meets all outstanding technical requirements.

Kind regards,

Kelvin Ian Afrashtehfar, M.Sc., D.D.S.,Dr. med. dent., FRCDC

Academic Editor

PLOS ONE

---

## [Editor Report · Acceptance letter]

24 Jan 2022

PONE-D-21-32339R1 

People’s perceptions of, willingness-to-take preventive remedies and their willingness-to-vaccinate during times of heightened health threats 

Dear Dr. Bearth:

I'm pleased to inform you that your manuscript has been deemed suitable for publication in PLOS ONE. Congratulations! Your manuscript is now with our production department. 

Kind regards, 

on behalf of

Dr. Kelvin Ian Afrashtehfar 

Academic Editor

PLOS ONE